# DeepGuiser: Learning to Disguise Neural Architectures for Impeding Adversarial Transfer Attacks

## Abstract

Security is becoming increasingly critical in deep learning applications. Recent researches demonstrate that NN models are vulnerable to adversarial attacks, which can mislead them with only small input perturbations. Moreover, adversaries who know the architecture of victim models can conduct more effective attacks. Unfortunately, the architectural knowledge can usually be stolen by the adversaries by exploiting the system-level hints through many side channels, which is referred to as the neural architecture extraction attack. Conventional countermeasures for neural architecture extraction can introduce large overhead, and different hardware platforms have diverse types of side-channel leakages such that many expert efforts are needed in developing hardware-specific countermeasures. In this paper, we propose DeepGuiser, an *automatic*, *hardware-agnostic*, and *retrain-free* neural architecture disguising method, to disguise the neural architectures to reduce the harm of neural architecture extraction attacks. In a nutshell, given a trained model, DeepGuiser outputs a deploy model that is functionally equivalent with the trained model but with a different (i.e., disguising) architecture. DeepGuiser can minimize the harm of the follow-up adversarial transfer attacks to the deploy model, even if the disguising architecture is completely stolen by the architecture extraction attack. Experiments demonstrate that DeepGuiser can effectively disguise diverse architectures and impede the adversarial transferability by $13.87\% \sim 32.59\%$, while only introducing $10\% \sim 40\%$ extra inference latency.

## 1 Introduction

Deep neural networks (NNs) have made great success in the field of artificial intelligence (AI) (LeCun et al., 2015). With NN becoming increasingly complex, a number of NN-specific chips (Jouppi et al., 2017; Liao et al., 2021; Markidis et al., 2018) and intensive innovations (Chen et al., 2020; Qiu et al., 2016; Chen et al., 2014) have been proposed to boost the efficiency of NN computing. Despite the significant progress in hardware performance, security should also be regarded as a higher-priority feature. Especially in safety-critical applications, e.g. autonomous driving, surveillance, and so forth, security vulnerabilities can be exploited by adversaries and lead to uncontrollable consequences.

Confidentiality is an essential guarantee for systemic security. The critical confidential information contained in well-trained NN models mainly includes their neural architectures and weight parameters. While the encryption of weight parameters has been well discussed for protecting the weight confidentiality (Orlandi et al., 2007; Cai et al., 2019; Zuo et al., 2021), the protection of neural architectures is still in lack. Recent researches have alerted that many emerging or even off-the-shelf AI chips are vulnerable to neural architecture extraction attacks (Batina et al., 2018; Hua et al., 2018; Yan et al., 2020; Hu et al., 2020; Wei et al., 2018; Wang et al., 2022). For example, DeepSniffer (Hu et al., 2020) exploits the system-level hints (e.g. memory access activity, cache miss rate, etc.) of NN processing on GPU platform and proposes a learning-based approach to automatically identify the layer sequences. It also quantitatively shows that the neural architecture extraction can significantly boost the success rate of adversarial transfer attacks by constructing a surrogate model with almost the same neural architecture as the victim model (Demontis et al., 2019; Hu et al., 2020).

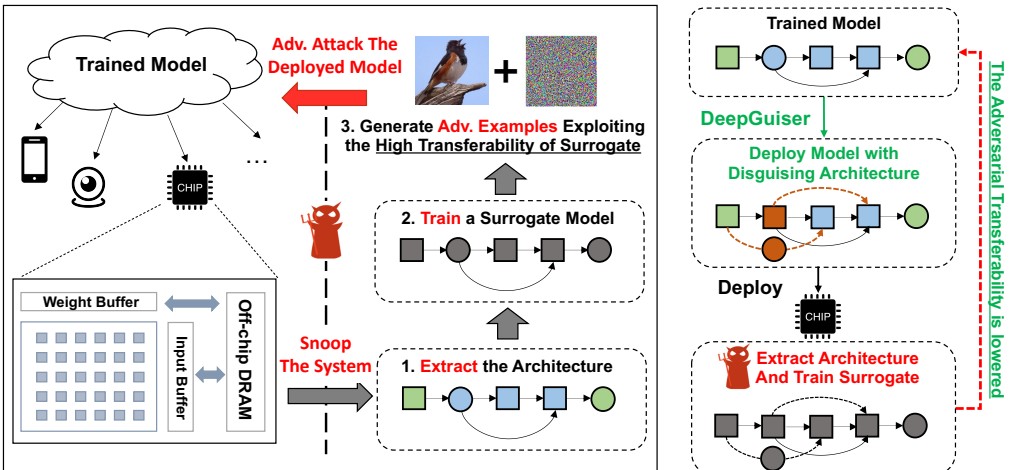

Figure 1: (Left) 1: The adversary can snoop the system to extract the architecture of the deployed model. 2&3: The architecture information can be utilized to train a surrogate with high transferability and then craft effective adversarial examples to attack the trained model. (Right) DeepGuiser disguises the trained model to a *functionally equivalent deploy model with a disguising architecture*. Then, this deploy model is deployed onto the chip. Even if the adversary extracts the disguising architecture through snooping and trains the surrogate model, the adversarial examples crafted using the surrogate have low transferability to the original trained model and also the actual deployed model.

The high risk rendered by neural architecture extraction attacks necessitates the protection of neural architectures. On one hand, from the view of intellectual property protection, neural architectures are usually manually designed by experts (He et al., 2016; Simonyan & Zisserman, 2014; Sandler et al., 2018) or automatically designed by neural architecture search (NAS) (Cai et al., 2018; Liu et al., 2018b; Tan et al., 2019), both of which consume significant labor and resources. On the other hand, from the view of adversarial robustness, if the architecture of the deploy model is leaked, the adversaries can train a surrogate model with the same architecture and use it to craft much more effective adversarial examples to attack the deploy model by exploiting the high transferability between the surrogate and deploy model (Hu et al., 2020).

It is hard to design a universal protection scheme against neural architecture extraction attacks for different kinds of AI chips at the system or hardware level, as various design options affect the hardware characteristics. Diverse run-time side-channel information can be exploited to extract the neural architectures on different hardware platforms, e.g. power (Wei et al., 2018), cache activity (Yan et al., 2020), memory access (Hua et al., 2018), etc. And blocking all these side-channel leakages by designing hardware-specific countermeasures might consume huge system costs and expert efforts.

In this work, we propose an "architecture disguising" solution at the algorithm level, DeepGuiser, which protects the architecture information by disguising it before deployment and alleviates the security risk rendered from the architecture extraction and the follow-up adversarial transfer attacks. Fig.1 (Left) illustrates the attack scenario we are concerned about, and Fig.1 (Right) demonstrates how DeepGuiser plays its role. And as there exist diverse models to be deployed and the disguising space (introduced in Sec. 4.1) is extremely large, manually finding a good disguising architecture for every possible model is extremely costly and even impossible. Therefore, we design DeepGuiser to automatically and efficiently yield a good disguising architecture for a given trained model. We summarize our contributions as follows:

- DeepGuiser is an **automatic**, **hardware-agnostic**, and **retrain-free** neural architecture disguising framework. As shown in Fig. 1 (Right), given a trained model, the disguising policy in DeepGuiser takes the original architecture as the input, and outputs a disguising architecture. Then, with functionality-preserving weight transforms, DeepGuiser yields a "deploy model" that is functionally equivalent with the trained model but with the disguising architecture. This deploy model is deployed, and even if its architecture is stolen, the harm of the follow-up adversarial transfer attacks to the deploy model can be largely reduced.

- We use reinforcement learning (RL) to train the disguising policy to output disguising architectures with low adversarial transferability to the original architecture. During the training process, we use a predictor for bridging the evaluation gap of the adversarial transferability between weight-sharing super-net and standalone training.

- For training the predictor, we build a dataset **TransAdvBench**, which collects and evaluates the adversarial transferability of over 8000 pairs of neural architectures. **TransAdvBench** can also help us study the connection between the architecture characteristics and adversarial transferability, and we list some of the knowledge in the **Appendix A.2**.

- Experimental results show that with DeepGuiser, the adversarial transferability of the surrogate model with the disguising architecture to the deploy model decreases by 13.87% $\sim$ 32.59%, while only introducing 10% $\sim$ 40% extra latency for the deploy model on GPU.

## 2 RELATED WORK

### 2.1 NEURAL ARCHITECTURE EXTRACTION ATTACKS

Many studies have proposed attack methods for extracting the neural architecture of deployed models on a variety of hardware platforms (Batina et al., 2018; Hua et al., 2018; Yan et al., 2020; Hu et al., 2020; Wei et al., 2018). For example, Hua et al. (2018) reveals the network architectures by utilizing the observed memory access pattern during the NN inference. DeepSniffer (Hu et al., 2020) proposes a learning-based operator recognition method by utilizing long short term memory (LSTM) network. Its basic idea is to learn the correlation between the system hints and the layer types. There are other side-channel information that can also be utilized to recognize the operations and topology, e.g. counting the GEMM calls via cache side-channel (Yan et al., 2020), observing the patterns and timing of operations (Batina et al., 2018), cache miss rate (Hu et al., 2020), etc. For example, scalar computation (e.g. ReLU) has higher cache miss rate compared to tensor computation (e.g. convolution) as the data reuse rate is much lower. These attack methods pose severe security risks for AI systems.

### 2.2 DEFENSIVE APPROACHES

There are two main streams of methods for defending against neural architecture attacks. One is to block the side channel leakage such that any adversary cannot obtain corresponding information. For example, memory access pattern and trajectory are important hints for helping the adversary reconstruct the network topology. The most promising solution, oblivious random-access machine (ORAM) protocol (Goldreich & Ostrovsky, 1996), can prevent the attacker from obtaining the actual access behavior. However, it is practically infeasible to apply ORAM owing to the unacceptable communication blowup. The most efficient implementation, Path ORAM (Stefanov et al., 2018), still has an overhead of $O(\log(N))$ blowup. Since data moving has already taken a significant proportion of time in NN computing, a great efficiency degradation will occur in bandwidth-limited chips.

Another stream of work is to obfuscate the neural architecture. However, the current methods are explicitly designed for defending against specific attack methodologies or with a high obfuscation cost. For example, NeurObfuscator (Li et al., 2021) is an obfuscation framework specifically targeting at increasing layer prediction error rate of the adversaries, without considering the ultimate criterion. ObfuNAS (Zhou et al., 2022) targets on hiding the accuracy performance, while requiring a searching process for every victim architecture.

### 2.3 NEURAL ARCHITECTURE SEARCH AND TRANSFORMATION

Neural architecture search (NAS) methods (Cai et al., 2018; Tan et al., 2019; Liu et al., 2018b) are widely studied to automatically design advanced architecture with superior performance in substitution of the hand-crafted design process. Essentially, NAS provides a convenient way for designers to explore a large architecture search space. Evaluation is one of the key components of NAS. To achieve fast and accurate estimation on the substantial architectures, predictor and architecture encoding methods are intensively discussed, e.g. GCN (Kipf & Welling, 2017; Guo et al., 2019), MLP (Liu et al., 2018a), GATES (Ning et al., 2020), etc. These schemes encode an architecture into a continuous embedding, which is used in the follow-up performance estimation.

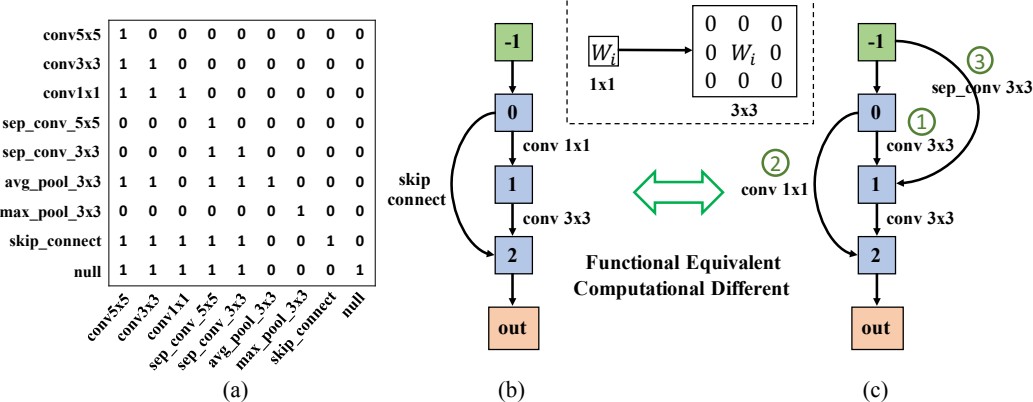

Figure 2: (a) The inclusion matrix of diverse operations in our implementation. The vertical axis represents original type and the horizontal axis represents the transformed type. In the table, "1" means corresponding disguising is valid and "0" means invalid; (b) an example of candidate neural architecture; (c) Disguised neural architecture from (b) based on the rules in (a). For example, transforming an 1x1 convolution kernel to 3x3 only needs to pad a surrounding zeros, then the two architectures are functionally equivalent while computationally different.

Neural architecture transformer (NAT) (Guo et al., 2019) aims to transform an architecture into a pruned one. It employs RL to train a policy that takes the original architecture as the input, and decides how to change each operation to get an architecture with fewer FLOPS and higher performance.

# 3 THREAT MODEL

The threat model considered is neural architecture extraction through hardware side-channel attacks, e.g., controlling the off-chip memory or snooping the communication bus. The adversary will obtain some hints that can help infer the deployed neural architectures through them. In edge computing scenarios, with AI becoming ubiquitous and mobile, more and more edge devices are powered with intelligence. An adversary can easily obtain physical access to the device by acquisition or theft. In a cloud computing scenario, an honest but curious cloud service provider may also seek the knowledge of NN models running on the cloud. They may fail to get the weight parameters due to encryption technologies (e.g., homomorphic encryption (Orlandi et al., 2007)), while it is much easier to reveal the model architectures because rich side-channel information can be observed.

We consider a strong threat model that the attackers can **exactly extract the neural architecture** of deployed models on some device while having **no ability to obtain the weight parameters** as the weight encryption techniques are relatively mature and strong. Then by training a surrogate model with the same neural architecture as the victim deploy model, the attackers can achieve a much higher adversarial attack success rate (Hu et al., 2020; Demontis et al., 2019). Under this scenario (illustrated in Fig.1 (Left)), the goal of neural architecture disguising is to find a disguised architecture that has lowest possible adversarial transferability to the actual deploy model.

# 4 METHOD

## 4.1 PROBLEM DEFINITION

Given an architecture $\mathcal{A}$, the objective of neural architecture disguising is to substitute a subset of its operations (layers) or add some operations (layers) to change its topology. Denoting $\mathcal{A}$ as a directed acyclic graph (DAG), it can be represented as $\mathcal{A} = (V, E)$, where $V$ is a set of nodes that represent the feature maps, and $E$ is a set of edges that represent the operations. For each operation $e_{ij}$, it computes on the feature map $v_i$ and produces feature map $v_j$. Assuming $e_{ij} = O_k$, an operation disguising is to change $O_k$ to $O_m$, satisfying that $O_m \in I(O_k)$. Here we define "$\in$" as the inclusion relationship, and $I(O_k)$ as the valid transformation set of operation $O_k$. Then, any operation transformation can only occur within its transformation set.

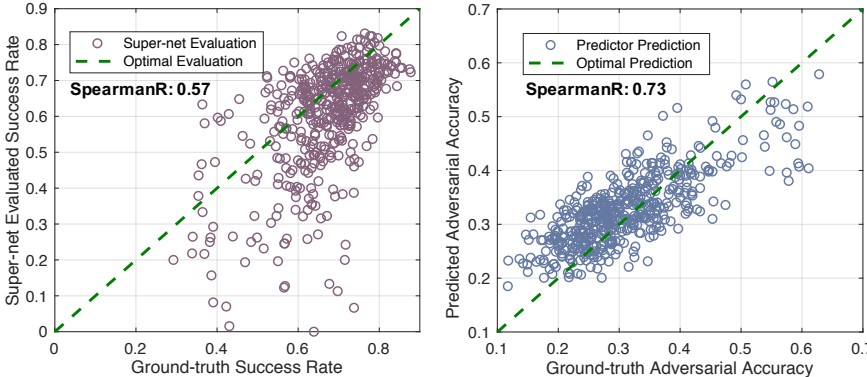

Figure 3: Adversarial transferability evaluation. Left: one-shot evaluation within super-net versus the ground-truth attack success rate; Right: evaluation with designed predictor versus the ground-truth transfer adversarial accuracy. The Spearman ranking correlation reflects the ranking quality of corresponding estimation.

**Disguising Space.** A principle of neural architecture disguising is maintaining equivalent functionality with the original model because functional correctness must be guaranteed. We can derive the disguising space according to this requirement, and design the corresponding functionality-preserving weight transforms for each operation disguising. For example, as shown in Fig.2(b-c), a 1x1 convolution can be disguised to a 3x3 convolution by padding its surrounding values as zero, then the 3x3 convolution can perform equivalent function as 1x1 convolution. Fig.2(a) concludes the disguising matrix of different operators. The matrix is applicable on any hardware because these rules are independent of the specific hardware implementation.

**Problem Formalization.** Given an architecture $\mathcal{A}$, the goal is to find an architecture $\mathcal{B}$ disguised from $\mathcal{A}$ to impede the adversarial transferability from $\mathcal{B}$ to $\mathcal{A}$. Denoting $\mathcal{A}_E = (e_1, e_2, ..., e_n)$ as the operation set of the architecture $\mathcal{A}$ and $\mathcal{B}_E = (e_{1d}, e_{2d}, ..., e_{nd}, e_{(n+1)d}, ..., e_{(n+m)d})$ as operation set of the architecture $\mathcal{B}$, the following constraint should be satisfied:

$$e_i \in \begin{cases} I(e_{id}), & 1 \leq i \leq n \\ I(null), & n < i \leq n + m \end{cases} \tag{1}$$

The objective is to minimize the loss of $\mathcal{A}$ on adversarial examples which are generated based on $\mathcal{B}$. Considering an adversarial example $x'_{\mathcal{B}}(x)$, it is usually formulated as:

$$x'_{\mathcal{B}}(x) = \operatorname{argmax}_z \mathcal{L}_{f_{\mathcal{B}}}(z, y), \quad s.t. ||z - x||_p \leq \epsilon, \tag{2}$$

where $f_{\mathcal{B}}(\cdot)$ denotes the forward function of entire model $\mathcal{B}$, x denotes the clean input, y denotes the label, z denotes the adversarial example, $\mathcal{L}$ denotes the loss function (often as utilized in training), and $\epsilon$ denotes the adversarial perturbation strength. $|| * ||_p$ denotes the $l_p$ norm, usually including $l_1$, $l_2$, and $l_\infty$, etc. Then the optimization problem for finding architecture $\mathcal{B}$ can be formalized as:

$$\operatorname{argmin}_{\mathcal{B}} \sum_{(x,y) \in \chi} \mathbb{E}_{(x,y)} \mathcal{L}_{f_{\mathcal{A}}}(x'_{\mathcal{B}}(x), y), \quad s.t. \mathcal{B} \in I(\mathcal{A}), \tag{3}$$

where $\chi$ represents the dataset, and $I(\cdot)$ represents the disguising space of some architectures. To measure the expectation, we use the boosted adversarial accuracy of $\mathcal{A}$ under the transfer attack of $\mathcal{B}$ as the reward, denoted as $R(\mathcal{B}|\mathcal{A})$. That is,

$$R(\mathcal{B}|\mathcal{A}) = AdvAcc_{\mathcal{B} \to \mathcal{A}} - AdvAcc_{\mathcal{A} \to \mathcal{A}}. \tag{4}$$

Unfortunately, the optimization problem is challenging to solve because of the problems from two aspects, i.e. evaluation and exploration.

**Evaluation Problem.** The evaluation of the reward is challenging. A standard evaluation requires a *training-generation-test* process, i.e. training a model with the disguised architecture $\mathcal{B}$, generating adversarial examples from model $\mathcal{B}$, and testing the adversarial accuracy of $\mathcal{A}$ by using the adversarial

examples to attack $\mathcal{A}$. Despite the process can give accurate transferability evaluation, all the steps (especially training) are time-consuming. The weight-sharing mechanism (Pham et al., 2018; Ning et al., 2021) is widely used in the NAS literature to accelerate the evaluation of architecture performances, i.e., to evaluate any architecture, the model directly uses the corresponding weights from a weight-sharing super-net. In this way, the above process can be simplified to a *generation-test* process by training a single super-net before the evaluation. However, we empirically find that the adversarial transferability evaluated by the super-net has a non-negligible gap with the ground-truth evaluations, as is shown in Fig.3(left). To address this problem, we propose a transferability predictor to fast and accurately estimate the adversarial performance, as will be introduced in Sec.4.2

**Exploration Problem.** Exploring the large disguising space also poses a challenge. Even with strict transformation constraints, the disguising spaces for most architectures is still extremely large. For example, a ResNet cell architecture shown in Fig.1 has approximately $(6^6 \times 2 \times 2)^2 > 10^{11}$ possible disguised architectures within the DARTS (Liu et al., 2018b) search space. Denoting $p(\cdot|\mathcal{A})$ as the probability distribution of sampling architecture $\mathcal{B}$ from the disguising space of $\mathcal{A}$, given the original architecture $\mathcal{A}$, we aim to find the disguising architecture distribution $p(\cdot|\mathcal{A})$ that maximizes $\mathbb{E}_{\mathcal{B}\sim p(\cdot|\mathcal{A})}[R(\mathcal{B}|\mathcal{A})]$. An intuitive idea is to learn a policy $\pi(\cdot|\mathcal{A})$ that takes $\mathcal{A}$ as the input and outputs the disguising architecture distribution for it. We model the disguising architecture distribution using the joint distribution of multiple operation disguising decisions. To learn this disguising policy, we employ policy gradient with our specific reward design $R$, as will be described in Sec.4.3.

The overall framework of DeepGuiser is shown in Fig.4(a). In the following, we will introduce the two main parts of DeepGuiser.

## 4.2 PREDICTOR: RESOLVING THE EVALUATION PROBLEM

To simultaneously achieve a fast and accurate evaluation on the adversarial transferability of any two architectures, we design a predictor to directly predict the adversarial accuracy given a victim architecture $\mathcal{A}$ and a surrogate architecture $\mathcal{B}$, i.e. the expectation $\mathbb{E}_{(x,y)}\mathcal{L}_{f_{\mathcal{A}}}(x'_{\mathcal{B}}(x), y)$.

**Predictor Construction.** The predictor consists of an architecture encoding block for transforming a discrete architecture into a continuous embedding space, and a regression head for predicting the adversarial accuracy from the architecture embedding. Specifically, we adopt a graph-based architecture encoder GATES (Ning et al., 2020) to convert an arbitrary architecture $\mathcal{A}$ to an embedding vector. Then, the embedding will be fed into an MLP-based regressor for regressing the adversarial accuracy, as shown in Fig.4(c).

**Loss Function.** The output of the predictor is a regression value which indicates the adversarial accuracy of architecture $\mathcal{A}$ over the adversarial examples generated by $\mathcal{B}$, we adopt the mean square error (MSE) loss to train the predictor.

$$\mathcal{L}(\theta_p, \mathcal{A}, \mathcal{B}, t) = (r(g(\mathcal{A}), g(\mathcal{B})|\theta_p) - t)^2, \tag{5}$$

where $r(\cdot)$ denotes the regressor over two input architecture embedding, $g(\cdot)$ denotes the architecture encoder GATES, $\theta_p$ denotes the parameters of the predictor, $t$ denotes the ground-truth adversarial accuracy of $\mathcal{A}$ under the attack from $\mathcal{B}$. Fig.3 (Right) shows the performance of the trained predictor. Compared to the super-net-based evaluation, the predictions are faster and more accurate.

## 4.3 POLICY LEARNING: RESOLVING THE EXPLORATION PROBLEM

We denote the disguiser function as $f_d$. Specifically, the disguiser parameters $\theta_d$ will be trained with the estimation of adversarial transferability given by the predictor.

**Disguiser construction.** Similar to the predictor, the disguiser also consists of an architecture encoding module that encodes an arbitrary architecture $\mathcal{A}$ to an embedding, and an MLP module that produces the policy $\pi(\cdot|\mathcal{A})$. The computation of the disguiser can be represented as:

$$\pi(\cdot|\mathcal{A}; \theta_d) = \text{Softmax}(f_d(g(\mathcal{A})|\theta_d)), \tag{6}$$

**Policy Gradient.** We apply policy gradient to learn the disguising policy. As the goal is to maximize the final reward $R(\mathcal{B}|\mathcal{A})$, the objective function can be formulated as:

$$\max_{\pi(\cdot|\mathcal{A})} \quad \mathbb{E}_{\mathcal{A}\sim p(\cdot)}[\mathbb{E}_{\mathcal{B}\sim \pi(\cdot|\mathcal{A};\theta_d)}[R(\mathcal{B}|\mathcal{A})]], \tag{7}$$

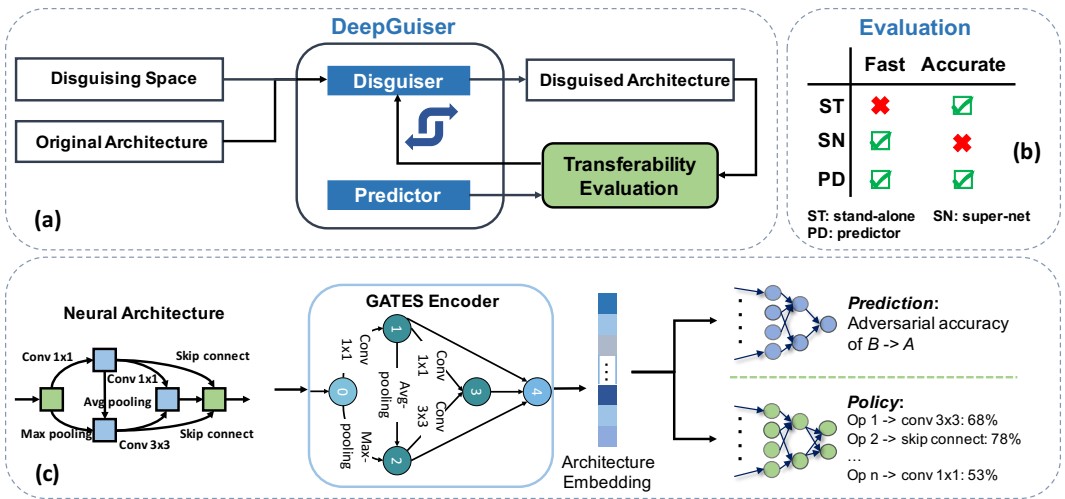

Figure 4: The framework and structure of DeepGuiser. (a) Overall flow of learning DeepGuiser; (b) Pros and cons of different evaluation methods; (c) the basic structure of predcitor and disguiser.

where $p(\cdot)$ is the probability distribution for sampling some architecture $\mathcal{A}$. In addition, we add two terms into the commonly-used loss function. Firstly, the operation disguising always introduces larger computational FLOPs and latency. Therefore, we add a penalty term on the reward corresponding to the transformed operation count, denoted as $c(\cdot|\mathcal{A})$. Secondly, we introduce a similar entropy regularization term $H(\pi(\cdot|\mathcal{A}; \theta_d))$ to encourage exploration. In summary, the objective function can be formulated as:

$$\mathcal{L}_{policy}(\theta_d) = \mathbb{E}_{\mathcal{A} \sim p(\cdot)}[\mathbb{E}_{\mathcal{B} \sim \pi(\cdot|\mathcal{A}; \theta_d)}[R(\mathcal{B}|\mathcal{A}) \cdot c(\mathcal{B}|\mathcal{A})] + \lambda H(\pi(\cdot|\mathcal{A}; \theta_d))],$$
$$= \sum_{\mathcal{A}} p(\mathcal{A})[\sum_{\mathcal{B}} \pi(\mathcal{B}|\mathcal{A}; \theta_d)(R(\mathcal{B}|\mathcal{A}) \cdot c(\mathcal{B}|\mathcal{A})) + \lambda H(\pi(\cdot|\mathcal{A}; \theta_d))]. \quad (8)$$

Specifically, in our implementation, the involved functions are formulated as follows respectively:

$$R(\mathcal{B}|\mathcal{A}) = r(g(\mathcal{A}), g(\mathcal{B})) - r(g(\mathcal{A}), g(\mathcal{A})), \quad c(\mathcal{B}|\mathcal{A}) = \frac{1}{2^{n_d - n} + 1}, \quad (9)$$

where $n_d$ is the number of disguised operations in the cell architecture (i.e. $n_d = \#\text{Diff}(\mathcal{A}, \mathcal{B})$), $n$ is a hyper-parameter that controls the penalty intensity. $H(\cdot)$ denotes the entropy of the distribution.

**Inference.** To infer a disguising for an architecture $\mathcal{A}$, we directly obtain the disguised architecture $\mathcal{B}$ by $\mathcal{B} = f_d(\mathcal{A}; \theta_d)$. Specifically, we obtain a probability distribution for each operation in the cell architecture, and the transformation with the maximum probability will be selected for disguising.

## 5 EXPERIMENTAL RESULTS

### 5.1 EXPERIMENT SETTINGS

**Benchmark.** We evaluate the effectiveness of DeepGuiser on a variety of neural architectures, including popular hand-crafted architectures (ResNet (He et al., 2016), VGG (Simonyan & Zisserman, 2014), and MobileNet-v2 (Sandler et al., 2018)) and randomly-picked architectures. We conduct experiments on three classification datasets, including CIFAR-10 (Krizhevsky et al., 2009), CIFAR-100 (Krizhevsky et al., 2009), and Tiny-ImageNet (Le & Yang, 2015) to demonstrate the generalization of adversarial transferability of specific architecture disguising. We choose projected gradient descent (PGD) (Madry et al., 2017) with 10 steps under the perturbation strength $\epsilon = 0.031$ (8/255) to generate adversarial examples and evaluate the adversarial accuracy. We also try C&W (Carlini & Wagner, 2017), AutoAttack (Croce & Hein, 2020), and DI-FGSM (Xie et al., 2019) adversarial example generation methods to evaluate the generalization on diverse attacks. For evaluating the

Table 1: Results of neural architecture disguising by different methods on CIFAR-10. $Acc_{adv}$ means the adversarial accuracy of the original model under black-box transfer attack from the disguised models. The Random-Arch and Random disguising report the average values of 20 randomly sampled architectures. The latency is tested on an NVIDIA GeForce RTX 3090 by averaging 1000 times of forward inference. "↑" represents higher better and "↓" represents lower better.

| Model | Method | #Params (M) ↓ | #FLOPS (M) ↓ | Latency (ms) ↓ | $Acc_{clean}$(%) ↓ | $Acc_{adv}$(%) ↑ |
|---|---|---|---|---|---|---|
| ResNet20 | / | 0.605 | 91.4 | 14.35 | 91.49 | 13.75 |
| | Random | 1.55 | 232.0 | 21.71 | 93.03 | 26.31 |
| | DeepGuiser-OS | **1.11** | **155.0** | 32.52 | 91.23 | 39.92 |
| | DeepGuiser | 1.22 | 182.4 | **20.46** | **89.06** | **46.34** |
| VGG16 | / | 0.605 | 91.4 | 13.29 | 88.82 | 24.89 |
| | Random | 1.52 | 227.7 | 21.93 | 92.62 | 26.83 |
| | DeepGuiser-OS | **1.08** | **152.6** | 29.55 | 92.33 | 31.26 |
| | DeepGuiser | 1.18 | 176.2 | **16.54** | **84.95** | **46.17** |
| MobileNetV2 | / | 0.257 | 46.2 | 15.08 | 88.39 | 15.31 |
| | Random | 1.16 | 180.6 | 23.18 | 92.97 | 30.05 |
| | DeepGuiser-OS | 0.801 | 119.1 | 30.33 | 92.35 | **41.80** |
| | DeepGuiser | **0.305** | **54.7** | **19.02** | **87.74** | 29.18 |
| Random-Arch | / | 0.812 | 122.3 | 15.9 | 90.19 | 33.23 |
| | Random | 1.98 | 276.1 | 22.7 | 92.43 | 44.96 |
| | DeepGuiser-OS | **1.36** | **196.5** | 27.7 | 91.70 | **51.38** |
| | DeepGuiser | 1.49 | 216.3 | **23.3** | **91.49** | 47.58 |

computational cost, we provide statistics of parameter size, FLOPS, and running latency of those architectures. The "DeepGuiser-OS" method represents the disguiser trained upon the one-shot evaluation given by weight-sharing super-net.

**Search Space.** We take DARTS search space (Liu et al., 2018b) as an example to evalaute the effectiveness of our method. Specifically, the cells are classified into normal cell and reduction cell. Both the cells contain 2 input node and 4 intermediate nodes. Every node can be the end of at most two edges. A total of 9 operation types are involved, as listed in Fig.2(a). For every conducted architecture, the base channel is set as 20 and the number of cells is set as 8.

**Structure.** The length of embedding vector produced by GATES is set as 128. Then the regressor MLP structure is 256x64x1 (as will concatenate the embedding of two architectures for prediction). The disguiser MLP structure is 128x256x512x144, where 144 equals (8 (operations in normal cell) + 8 (operations in reduction cell)) $\times$ 9 (possible disguising options).

**Training Details.** The training of DeepGuiser contains two parts. For the training of predictor, we utilize the constructed dataset *TransAdvBench* built upon the DARTS search space, including 8,082 pairs of neural architectures as train data and 484 pairs as test data (see **Appendix A.1** for more details). We train the predictor by setting the learning rate as 0.001, batch size as 64, epochs as 30, and weight decay as 0.0005. For the training of disguiser, we set the learning rate as 3e-4, iteration number as $10^4$, entropy coefficient $\lambda$ as 0.003, penalty controller $n$ as 10.

## 5.2 THE RESULTS OF IMPEDING ADVERSARIAL TRANSFERABILITY

Table 1 concludes the key metrics for evaluating the effectiveness and efficiency of neural architecture disguising. As can be seen, DeepGuiser can find a better policy for disguising diverse architectures to defend against architecture extraction attacks. Compared to constructing a surrogate model with the same architecture as the victim model, when an adversary can only obtain the disguised fake architecture, the attack success rate will significantly drop, e.g. 33.55% higher adversarial accuracy will be maintained for ResNet20 when it is attacked by the disguised ResNet20. We also evaluate the cost for disguising those architectures. Inevitably, disguising will introduce extra computation and parameters. The experiments show that DeepGuiser outperforms the other methods at even a lower latency cost. Moreover, we surprisingly find that the clean accuracy of disguised architectures by DeepGuiser also significantly drops, further reducing the profits of the adversaries from the attacks, i.e. the attacker obtains much worse architectures while the actual architectures are hidden.

Table 2: Results of neural architecture disguising under different attack methods. All numbers are the adversarial accuracy of the original model under black-box transfer attack from the disguised models.

| Model | Method | AutoAttack | C&W | DI-FGSM | Average |
|---|---|---|---|---|---|
| ResNet20 | / | 4.2 | 20.21 | 1.1 | 8.5 |
| | Random | 15.49 | 23.77 | 9.45 | 16.24 |
| | DeepGuiser-OS | 17.46 | 24.8 | 10.28 | 17.51 |
| | DeepGuiser | 15.3 | 30.58 | 11.16 | **19.01** |
| VGG16 | / | 8.2 | 22.98 | 3.29 | 11.49 |
| | Random | 11.73 | 25.03 | 8.73 | 15.16 |
| | DeepGuiser-OS | 24.08 | 27.94 | 16.98 | 23 |
| | DeepGuiser | 22.83 | 31.52 | 15.07 | **23.14** |
| MobileNetV2 | / | 5.5 | 22.39 | 2.13 | 10.01 |
| | Random | 14.96 | 24.25 | 12.02 | 17.08 |
| | DeepGuiser-OS | 23.18 | 24.17 | 18.48 | **21.94** |
| | DeepGuiser | 8.68 | 27.8 | 7.92 | 14.8 |

Table 3: Results of neural architecture disguising by different methods on CIFAR-100 and Tiny-ImageNet. $Acc_{adv}$ means the adversarial accuracy of the original model under black-box transfer attack from the disguised models. "↑" represents higher better and "↓" represents lower better.

| Model | Method | CIFAR-100 | | | Tiny-ImageNet | | |
|---|---|---|---|---|---|---|---|
| | | Latency (ms) ↓ | $Acc_{clean}$ (%) ↓ | $Acc_{adv}$ (%) ↑ | Latency (ms) ↓ | $Acc_{clean}$ (%) ↓ | $Acc_{adv}$ (%) ↑ |
| ResNet20 | / | 12.41 | 68.09 | 12.54 | 12.73 | 44.92 | 14.46 |
| | DeepGuiser-OS | 27.12 | 68.54 | **42.84** | 26.20 | 53.28 | 23.38 |
| | DeepGuiser | **18.01** | **63.86** | 40.70 | **18.66** | 47.62 | **23.72** |
| VGG16 | / | 13.25 | 62.39 | 24.77 | 13.96 | 40.76 | 21.65 |
| | DeepGuiser-OS | 24.63 | 68.36 | 36.92 | 24.44 | 54.52 | 23.53 |
| | DeepGuiser | **23.23** | **53.07** | **39.87** | **15.96** | **41.23** | **25.25** |
| MobileNetV2 | / | 15.54 | 61.88 | 28.31 | 15.97 | 41.26 | 7.58 |
| | DeepGuiser-OS | 25.21 | 69.74 | 36.71 | 26.41 | 53.92 | **24.64** |
| | DeepGuiser | **17.55** | **60.42** | **37.01** | **17.51** | 44.29 | 16.33 |

## 5.3 GENERALIZATION ON DIFFERENT ATTACK METHODS

In this experiment, we evaluate the generalization of adversarial transferability under different attack methods. Table 2 concludes the results. As can be seen, the adversarial transferability from surrogate models to victim models can still be impeded (though not as significant as under PGD attack) with different adversarial example generation methods. This eliminates the need to build a dataset corresponding to each attack method and perform repeated training for all types of attacks.

## 5.4 GENERALIZATION OF THE DISGUISING POLICY TO OTHER DATASETS

We explore whether the disguising policy trained on CIFAR-10 can be utilized on other datasets. We test the adversarial transferability of the disguising architecture to the original architecture on CIFAR-100 and Tiny-ImageNet. As shown in Table 3, the disguised architectures still demonstrate consistent results as in the experiments on CIFAR-10. The results suggest that the transferability between architectures is largely determined by their architecture characteristics and general on different datasets. That is to say, DeepGuiser can be trained on a proxy dataset like CIFAR-10 and generalized to other datasets, eliminating the need to collect TransAdvBench for every new dataset.

## 6 CONCLUSION

In this work, we propose DeepGuiser to automatically disguise neural architectures for impeding the adversarial transfer attacks after neural architecture extraction. DeepGuiser employs a predictor that predicts the adversarial transferability between architectures to learn a disguising policy to transform the architecture operations. DeepGuiser converts the trained model to a functionally-equivalent "deploy model" with a disguising architecture in a hardware-agnostic and post-training way, which can be applied no matter what hardware platform is used and incorporates no retraining or fine-tuning of the trained model. Experimental results show that DeepGuiser can effectively impede the effectiveness of adversarial transfer attacks following the architecture extraction attack.

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

## A  APPENDIX

### A.1  TRANSADVBENCH

As illustrated in the main manuscript, TransAdvBench is built for benchmarking the adversarial transferability between neural architectures. Here we present the details on the collection, annotation, statistics, and quality of the dataset.

**Data Source.** To make the samples more representative and generalized, we collect diverse neural architectures from the DARTS search space. Every architecture will be built upon a specific normal cell architecture and a specific reduction cell architecture. Both the cells contain 2 input node and 4 intermediate nodes. Every node can be the end of at most two edges. A total of 9 operation types are involved. The base channel number is set as 20 and every architecture will cascade eight cells.

**Data Construction.** Every data point will contain two architectures, one for victim neural architecture (denoted as $\mathcal{A}$), and the other for surrogate neural architecture (denoted as $\mathcal{B}$) to generate adversarial examples for transfer attack. The annotations for every architecture pair include: clean accuracy of $\mathcal{A}$, clean accuracy of $\mathcal{B}$, adversarial accuracy of $\mathcal{A}$ under the transfer attack from $\mathcal{B}$.

**Data Collection.** To collect a data point, we fully train the two neural architectures independently. We split the full train set of CIFAR-10, which has 50,000 images in total, then take 80% of the data for training and 20% of the data for validation, to ensure the strict isolation of train data and validation data. Every model will be fully trained on the train subset and tested on the validation subset. We choose projected gradient descent (PGD) with 10 steps under the perturbation strength $\epsilon = 0.031$ (8/255) to generate adversarial examples and evaluate the adversarial accuracy. Every single model is trained with a batch-size of 64, a cosine learning rate scheduling with maximum 0.05 and minimum 0.01, and a total of 50 epochs.

**Statistics.** Here we provide some statistics of the dataset. Overall, TransAdvBench contains 8,082 data points for training and 484 data points for testing. The 8,082 train data come from 5,473 different neural architectures and the 484 data point come from 605 different neural architectures. Among the training dataset, we pick 1,117 different architectures to be the victim architectures, and every architecture will have four surrogate architectures to produce the data points. That is, $1,117 \times 4 = 4,468$ data points are produced. Then we shuffle all architectures and randomly select another 3,614 victim-surrogate pairs to produce the left data points. As the main purpose of this work is to identify the transferability between an original architecture and its disguised architectures, we construct the test dataset by sampling 121 victim architectures. For each victim architecture, 4 other surrogate architectures will be sampled based on the disguising rules.In total, there will be $121 \times 4 = 484$ data points in test set. Note that we strictly ensure the isolation of the neural architectures in train set and test set.

Fig.5(Left) shows the statistical distribution of the clean accuracy of sampled neural architectures. Most of the data distribute on the range of 0.9 to 0.97. Fig.5(Right) shows the statistical distribution of the adversarial accuracy of the data points. It can be figured out that the adversarial accuracy spreads over a wide range, while the distribution is non-uniform and has long tails, thus might cause biasing to the predictor training. Nevertheless, the training data points can provide sufficient generalization on diverse architectures, and can be used for guiding the training of predictor.

To our best knowledge, this is the first data bench for evaluating the adversarial transferability among diverse neural architectures. Since the collection of each data point is expensive (requires a *training-generation-test* process), the scale of the dataset is still limited currently , and we only take PGD-10 with perturbation strength 0.031 for evaluating the adversarial accuracy. We are working on building a larger dataset with more reasonable sampling method and richer annotations.

### A.2  INSIGHTS FROM TRANSADVBENCH

We provide several insights about the connection between adversarial transferability and neural architectures. To be specific, we attempted to answer two questions: What are the features of a easy-to-disguise neural architecture? What kinds of operation transformations can effectively reduce the adversarial transferability between neural architectures? Since TransAdvBench gives accurate transferability evaluation between thousands of neural architectures, we can get relatively credible

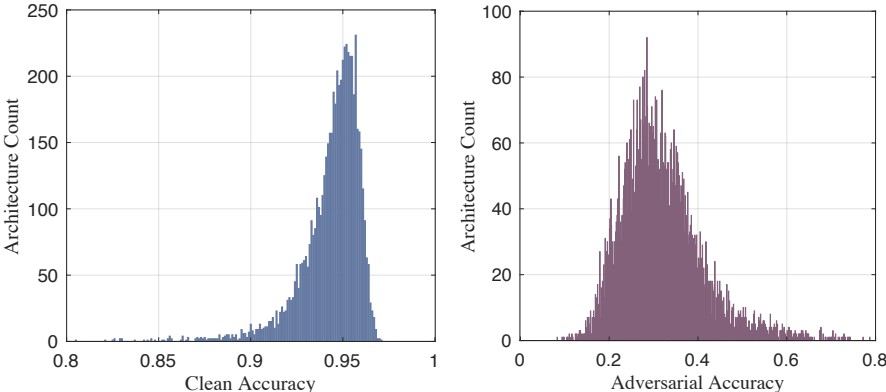

Figure 5: Statistics of TransAdvBench. Left: the distribution of clean accuracy of the sampled neural architectures. Right: the distribution of the adversarial accuracy of the 8,566 (8,082 train data plus 484 test data) pairs of neural architectures.

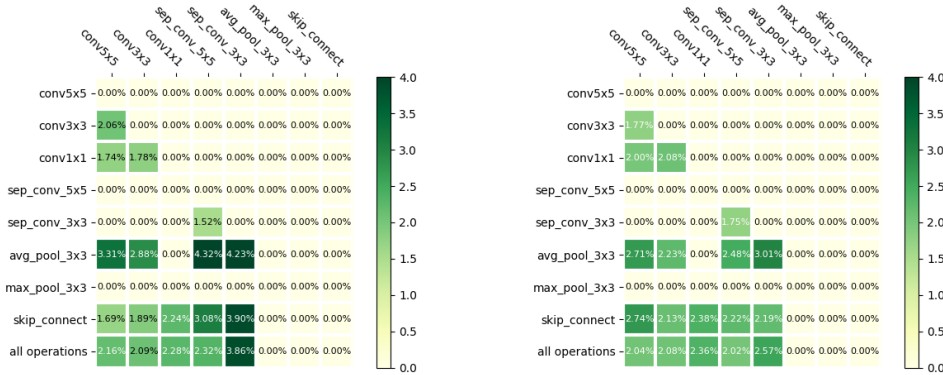

Figure 6: Average adversarial accuracy increase of each operation transformation. The numbers in the figure are the average adversarial accuracy increase on each operation transformation. For example, assume in 10 victim and surrogate architectures at least one skip connection is transformed to convolution kernels with size 5x5, the total adversarial accuracy increase of the 10 pairs are 20%. The the skip_connect/conv_3x3 in the table should be 20% / 10 = 2%. Left: Average adversarial accuracy increase of normal cells; Right: Average adversarial accuracy increase of reduction cells. The bottom row indicates the average reward transformed from all operations to a specific operation. For example, assume in 10 surrogate architectures there exist at least one sep_conv_3x3 transformed from other operations. The total adversarial accuracy increase of the 10 pairs is 20%, then all operations/sep_conv_3x3 in the table should be 2%

insights from analyzing TransAdvBench. We show the statistics on average adversarial accuracy increase in Fig.6 and the statistics on the portion of operations in victim architectures in Fig.7. The statistics involves 1760 victim and surrogate architectures in TransAdvBench in which operation transformations strictly meet the disguising rules. From the statistics we get the following insights.

**Insight 1**: Disguising normal cells leads to higher average adversarial accuracy increase than disguising reduction cells. From Fig.6 we notice that the biggest average adversarial accuracy increase by transforming operation in normal cell is 4.32% while in reduction cell 3.01%. Additionally, the operation transformations with top 5 average adversarial accuracy increase are all in normal cells. One possible reason for this phenomenon is that in the disguising space of our work, a neural architecture is completed by repeatedly assembling the normal cell architecture and the reduction cell architecture

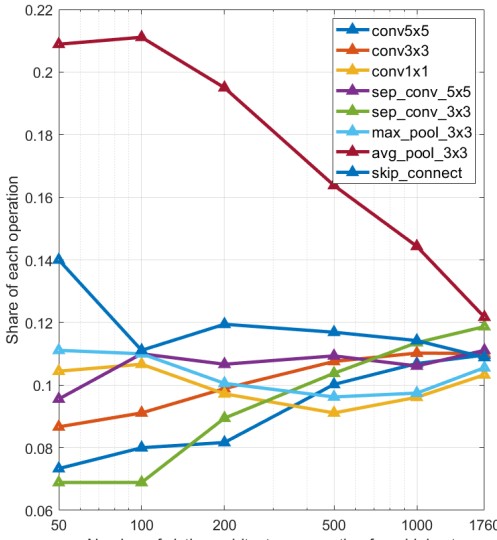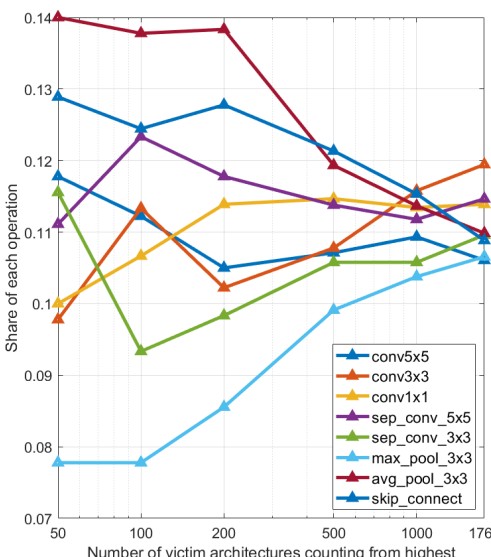

Figure 7: Average share of each operations in victim architectures. We sort the 1760 victim and surrogate architectures by their adversarial accuracy increase from high to low. The x-axis indicates the number of architectures (counting from highest) are used to compute the average share. The y-axis indicates the average share of an operation in victim architectures. For example, (50, 0.21) on the avg_pool_3x3 line means that 21% of operations in victim architectures whose surrogate architectures yield the highest 50 adversarial accuracy increase are avg_pool_3x3. Left: Average share of normal cells; Right: Average share of reduction cells.

in sequence. Normal cell architecture is repeated more times than reduction cells in the complete neural architecture thus is more crucial in operation transformation.

**Insight 2**: A strong correlation exists between number of average pooling operations in a victim architecture and adversarial accuracy increase.The victim architectures in TransAdvBench are randomly sampled so the number of each operations in TransAdvBench are generally equal. Interestingly enough, as is shown in Fig.7 by the avg_pool_3x3 line,along with the decrease of number of victim architectures (counting from highest) ,the share of avg_pool_3x3 in victim architectures continuously grows. In victim architectures with top 50 adversarial accuracy increase 21% normal cell operations are avg_pool_3x3. In contrast only 12% normal cell operations in the entire 1760 victim and surrogate architectures are avg_pool_3x3.

**Insight 3**: Transforming an operation to sep_conv_3x3 yields the biggest adversarial accuracy increase. The operation transformations in TransAdvBench are completely random. From the bottom row of Fig.6 we can see that transforming an operation to sep_conv_3x3 yields 3.86% adversarial accuracy increase in normal cell and 2.57% adversarial accuracy increase in reduction cell, both exceed other operations.

## A.3 TRAINING CURVES

Fig.8 shows the curves of predictor training based on MSE loss presented as Equation 5 in the main manuscript. For making the loss more significant, we multiply a scaling factor 100 for the loss values. As can be seen, the predictor converges fast, with a rapid loss value dropping and a significant increasing on the Kendall's Tau of the predictions. Moreover, the trained predictor can generalize to test set, achieving superior prediction on both adversarial accuracy and the ranking quality. The experimental results show that adversarial accuracy of some architecture $\mathcal{A}$ under the adversarial transfer attack from another architecture $\mathcal{B}$ is predictable, suggesting that there are internal characteristics in neural architectures that affecting the adversarial transferability. Therefore, capturing the characteristics is feasible such that we can learn an automatic disguiser to discover the least transferable architecture to any victim architecture.

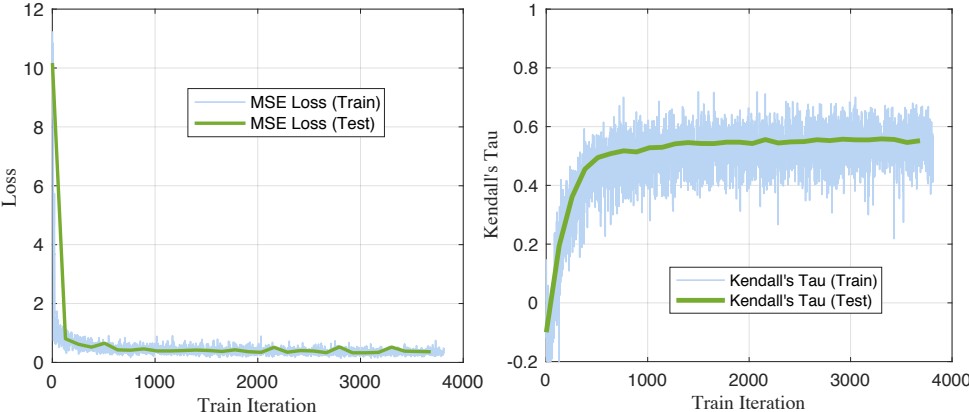

Figure 8: The training curves of predictor. Left: MSE loss curves of training and testing; Right: the Kendall's Tau of training and testing.

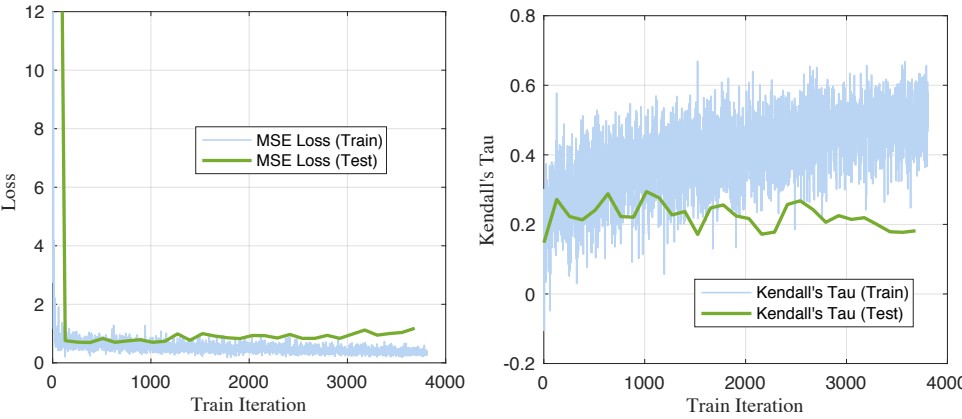

Figure 9: The training curves of predictor based on GCN encoder. Left: MSE loss curves of training and testing; Right: the Kendall's Tau of training and testing.

## A.4 ABLATION STUDY ON THE ARCHITECTURE ENCODING SCHEMES

We further compare the performance of different architecture encoding schemes, including GATES method and GCN method (utilized by NAT). As shown in Fig.9, when applying GCN for generating the architecture embedding, the predictor tends to overfit to the train dataset, with a much higher MSE loss and a much lower Kendall's Tau on test dataset compared to GATES-based encoding. The results demonstrate the superiority of GATES as the architecture encoder as it provides better modeling of the neural architectures and can help the predictor to obtain better ranking quality and lower prediction error.

## A.5 DISGUISING PROBABILITY STATISTICS

We show the statistics on the probability of operation transformations in Fig.10. Several insights can be figured out. First, DeepGuiser tends to disguise the convolutional layers with expanded kernels, e.g. conv3x3 has a high probability of 87% to be disguised as conv5x5, and conv1x1 has a probability of 77% to be disguised to conv3x3 or conv5x5. Second, changing the network topology is expected to achieve higher reward, as skip connection has a higher probability of 92% to be disguised to other operations and null also has a probability of 66% to be changed. Overall, the probability distributions

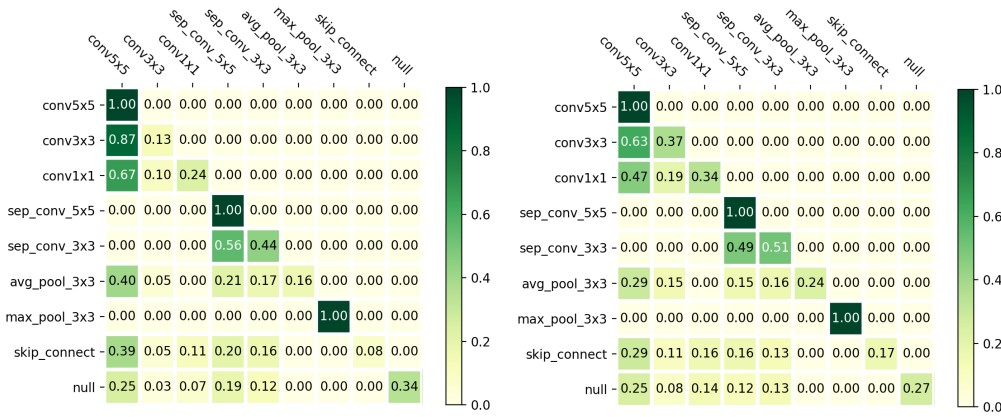

Figure 10: Probability of operation transformations. 1000 different architectures are sampled to count the probability. Left: probability of normal cells; Right: probability of reduction cells. The zeros numbers means that the disguising is not allowed due to computational rules.

of most operations are relatively balanced, guaranteeing the diversity of neural architecture disguising. If the disguising patterns are fixed, it will reduce the concealment of neural architecture disguising if the adversaries master the disguising pattern.

## A.6 VISUALIZATION

We post the graph views of several neural architecture disguising samples in Fig.11,12,13. It is quite difficult for the attackers to guess the disguised operations from observation, thereby preventing the disguising from being easily cracked by the adversaries.

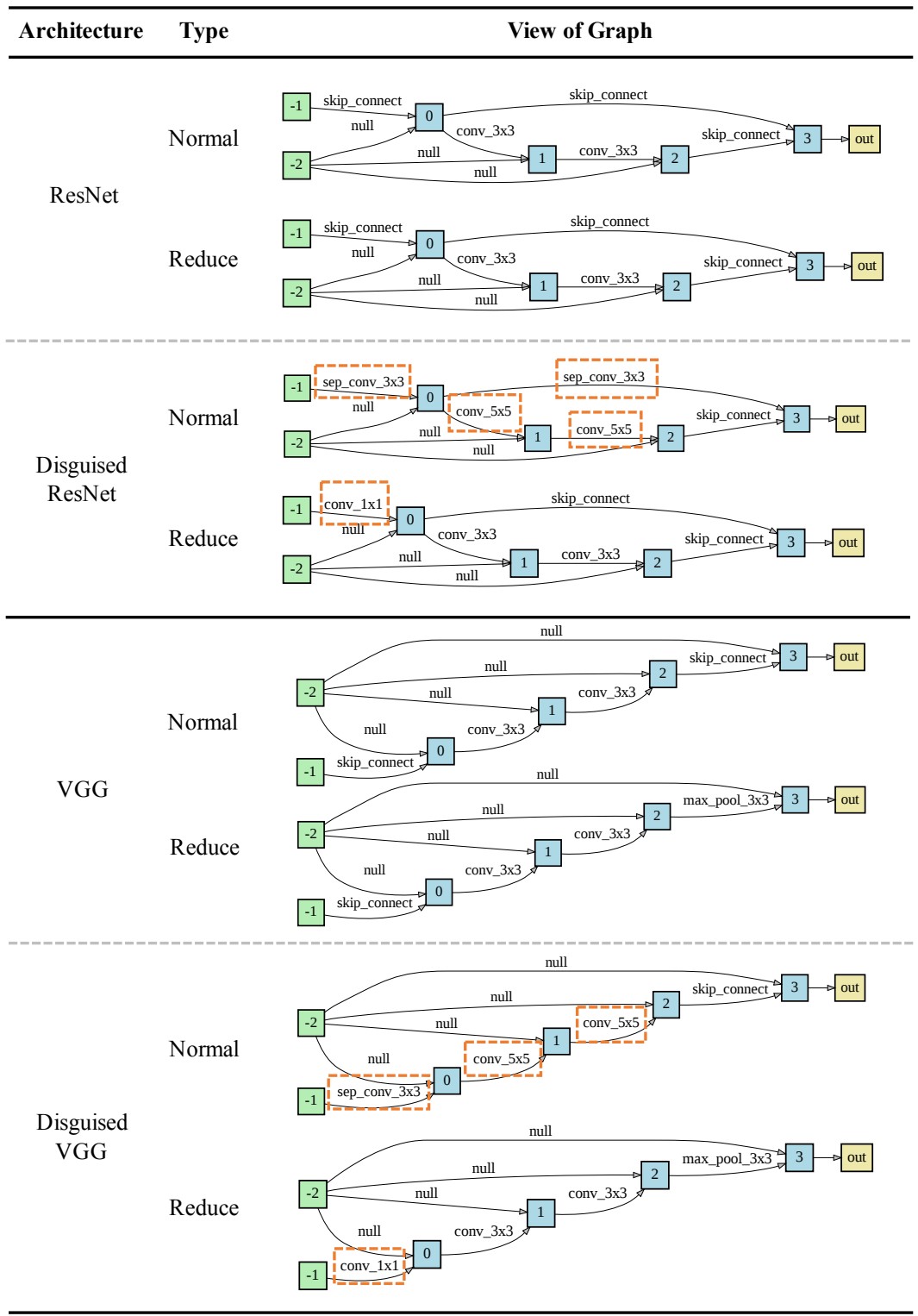

Figure 11: Graph view of sampled neural architectures (I).

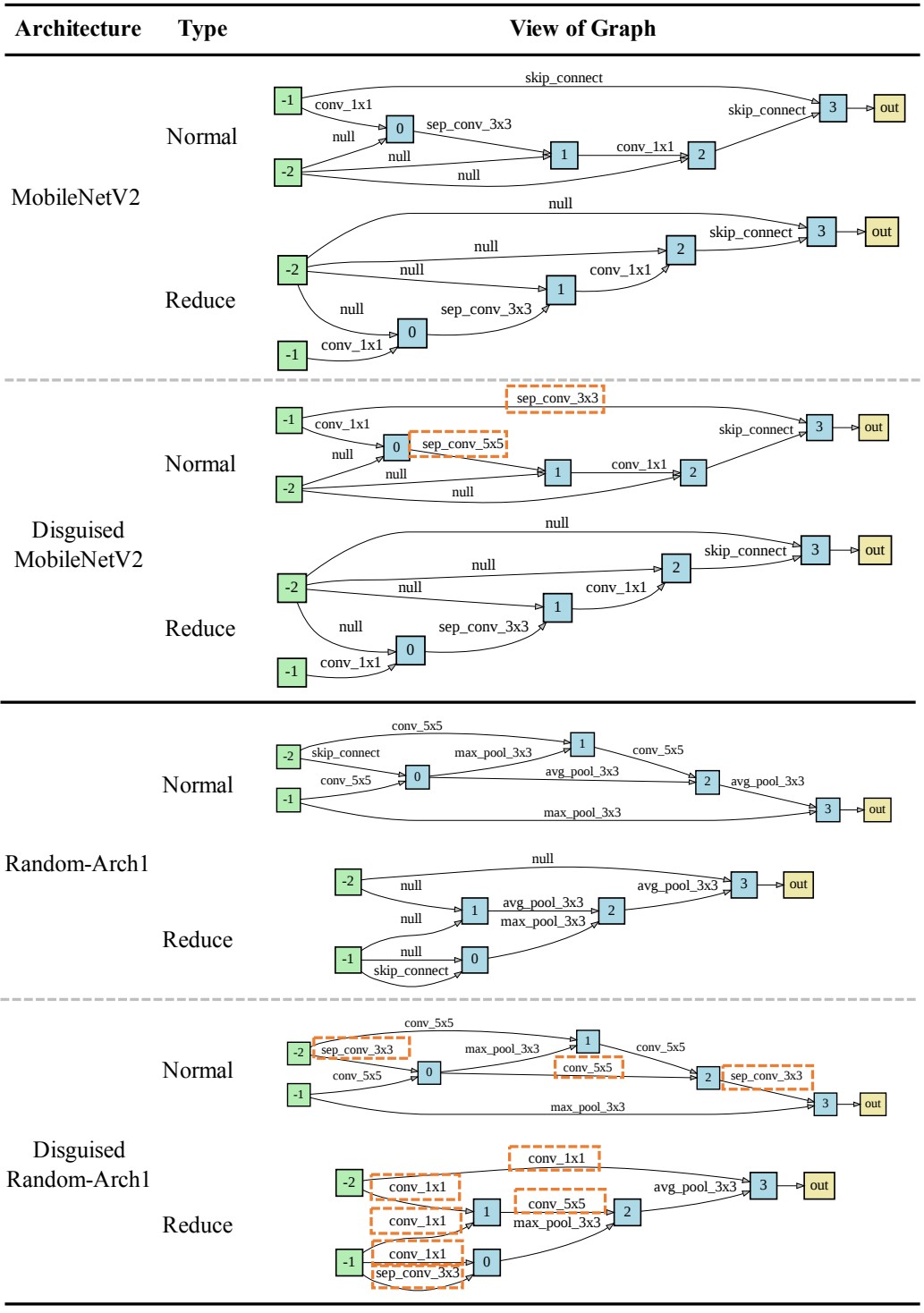

Figure 12: Graph view of sampled neural architectures (II).

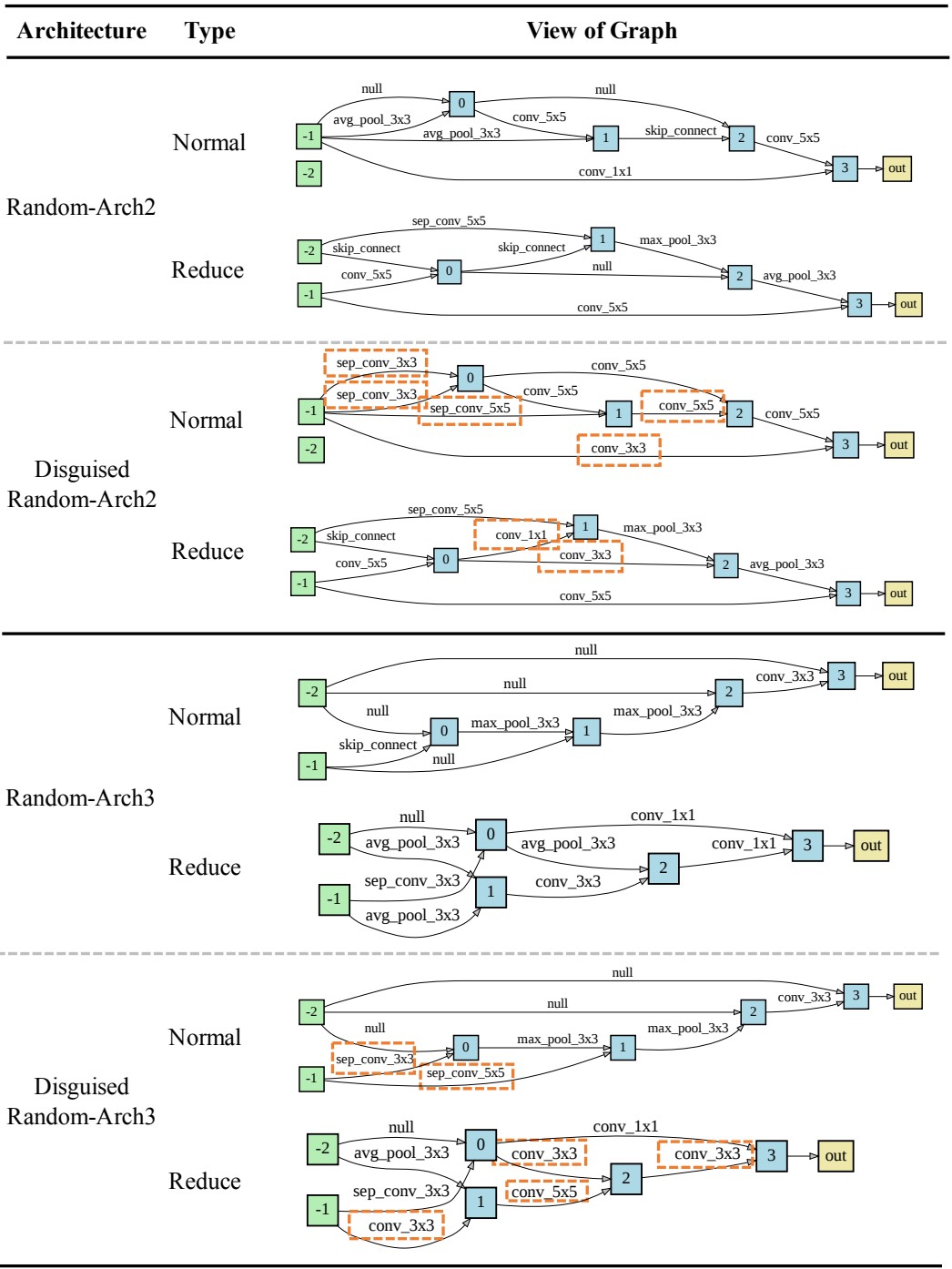

Figure 13: Graph view of sampled neural architectures (III).

