# OpenReview forum: "DeepGuiser: Learning to Disguise Neural Architectures for Impeding Adversarial Transfer Attacks"
_ICLR.cc/2023/Conference — Submitted to ICLR 2023_

### Official Review · Reviewer_Tj1c · 2022-10-17

**Confidence:** 5
**Correctness:** 2
**Technical Novelty And Significance:** 3
**Empirical Novelty And Significance:** 3
**Recommendation:** 3

**Clarity, Quality, Novelty And Reproducibility:**

The expression of this paper is relatively clear and the method has some novelties, It should be able to reproduce well.

**Strength And Weaknesses:**

Strength:
1. It is a potential idea to camouflage the neural network structure to reduce the success rate of the transfer attack.
2. When disguising the neural network structure, the clean accuracy remains almost unchanged.

Weaknesses:
1. The paper says to defend against transfer attacks, but it does not compare state-of-the-art migration attacks, such as MI-FGSM [1], TI-FGSM [2], PI-FGSM [3] and VMI-FGSM [4]. Both AutoAttack and C&W are common white-box attacks.
2. The assumptions of this scenario are too strict and may limit the generalizability of the method.
3. In this setting, the defense performance of the method appears to be inferior to the adversarially trained model. The latter not only defends against transfer attacks, but is also effective against white-box attacks.
4. The experimental model (ResNet20) is not common enough, can you consider ResNet50 or Wide-ResNet?

[1] Dong, Yinpeng, Fangzhou Liao, Tianyu Pang, Hang Su, Jun Zhu, Xiaolin Hu, and Jianguo Li. "Boosting adversarial attacks with momentum." In Proceedings of the IEEE conference on computer vision and pattern recognition, pp. 9185-9193. 2018.
[2] Dong, Yinpeng, Tianyu Pang, Hang Su, and Jun Zhu. "Evading defenses to transferable adversarial examples by translation-invariant attacks." In Proceedings of the IEEE/CVF Conference on Computer Vision and Pattern Recognition, pp. 4312-4321. 2019.
[3] Gao, Lianli, Qilong Zhang, Jingkuan Song, Xianglong Liu, and Heng Tao Shen. "Patch-wise attack for fooling deep neural network." In European Conference on Computer Vision, pp. 307-322. Springer, Cham, 2020.
[4] Wang, Xiaosen, and Kun He. "Enhancing the transferability of adversarial attacks through variance tuning." In Proceedings of the IEEE/CVF Conference on Computer Vision and Pattern Recognition, pp. 1924-1933. 2021.



**Summary Of The Paper:**

In this paper, the authors discuss how to defend against adversarial transfer attacks when the model structure is leaked. Considering that weight encryption techniques are relatively mature and strong, the authors consider the threat model that an attacker can exactly extract the neural architecture of deployed models on a device but not the weight parameters. Therefore, the author proposes DeepGuiser converts the trained model to a functionally-equivalent “deploy model” with a disguising architecture in a hardware-agnostic and post-training way, which can be applied no matter what hardware platform is used and incorporates no retraining or fine tuning of the trained model. According to experimental results, DeepGuiser can effectively impede adversarial transfer attacks.

**Summary Of The Review:**

We tend to reject this paper because of its more restrictive assumptions and lack of more convincing experiments.

---

> ### Author Response · Authors · 2022-11-19
> **Responses to Reviewer Tj1c**
>
> We thank the reviewer for the constructive suggestions and comments to help us improve our work. The following lists our responses.
>
> Q1: compare state-of-the-art migration attacks
>
> A: We will add the mentioned attacks into our evaluation in the next revision. We think the experimental phenomena will keep aligned with current conclusion, as PGD is quite similar to FGSM-based variants (MI-FGSM, TI-FGSM, PI-FGSM, etc.)
>
> Q2: The assumptions of this scenario are too strict and may limit the generalizability of the method.
>
> A: We think the attack scenario is becoming increasingly general in the future as AI chips are entering more and more applications from cloud to edge. Moreover, the DeepGuiser framework is not limited to impeding adversarial example attacks. One can also design other criteria and reward function to protect what they think is important, e.g. model accuracy, layer predictor error rate, etc.
>
> Q3: In this setting, the defense performance of the method appears to be inferior to the adversarially trained model.
>
> A: We agree with you that adversarial training is a much stronger method that can directly enhance the model robustness against adversarial attacks. However, it requires a time-consuming adversarial training process, as every iteration we need to generate adversarial examples on-the-fly. And adv train usually has negative impact on the model clean accuracy. Our method is a post-training disguising, without any re-training or adding any overhead to the model training process. Moreover, our method is orthogonal to adv train, such that they can be integrated to make the transfer attack less effective.
>
> Q4: The experimental model (ResNet20) is not common enough, can you consider ResNet50 or Wide-ResNet?
>
> A: ResNet50 is also built upon ResBlock, and Wide-ResNet has wider channels while also has the same architecture as standard ResBlock. We cannot deliver complex results (e.g. ResNet-50 over ImageNet) during rebuttal period and we will add more comprehensive experiments in the next revision.

---

> > ### Comment · Reviewer_Tj1c · 2022-12-01
> > **Acknowledgement of Rebuttal**
> >
> > I would like to thank the authors for the detailed feedback.
> > After reading the response, I still feel that the paper is not ready for publication.
> > 1. There are no additional results to address my concerns.
> > 2. From my knowledge, PGD is not similar to FGSM-based variants (MI-FGSM, TI-FGSM, PI-FGSM, etc.)
> >
> > I keep my score.

---

### Official Review · Reviewer_PPSy · 2022-10-18

**Confidence:** 4
**Clarity, Quality, Novelty And Reproducibility:** See Strength And Weaknesses.
**Correctness:** 3
**Technical Novelty And Significance:** 4
**Empirical Novelty And Significance:** 4
**Recommendation:** 5

**Strength And Weaknesses:**

Strength
+ The idea to deploy a different architecture with the exactly same operation results is novel, and naturally hardware-agnostic and retrain-free.
+ The method to minimize the adversarial transferability is technically sound with careful design.
+ The TransAdvBench contributes to the community in research of attack transferability.
+ Solid experiments validate the superiority of the proposed method in mitigating transfer attacks.
+ The paper is well-motivated and easy to follow with available codes.

Weakness
+ The architecture is not the only factor affecting adversarial transferability. Although it is an important factor, the author needs to answer whether “the same pair of architecture with different parameters have similar transferability”. If modifying parameters (but maintaining the architecture) could greatly change the ranking of TransAdvBench, then its significance would be questioned.
+ The proposed method largely increases the inference cost. Although it is understandable that great changes in the architecture tend to decrease the transferability more, transfer attacks via architecture extraction attacks may not be that popular in real-world scenarios. A better choice is to consider the model size or FLOPs when searching the architecture, enabling the deployer to trade-off.
+ Does the Acc_clean in Tables 1 & 3 stand for clean test accuracy? If so, why is lower accuracy better, and why does a CIFAR-10 baseline have only 91.5 accuracy (which should be around 95, the same problem exists in CIFAR-100 and Tiny-ImageNet)? If not, what is it? Also, the related bolding is confusing and incorrect.


**Summary Of The Paper:**

This paper proposes a defense against neural architecture extraction attacks, which would effectively hurt the model through transfer attacks. The presented method seeks a replacement architecture that has the same operation results as the original model but with low adversarial transferability, which is estimated by a transferability predictor with policy gradient optimization. The TransAdvBench used to pre-train the predictor also constructs a new benchmark.

**Summary Of The Review:**

It is a good and original paper on motivation and design. But the experiments have some issues needed to be addressed.

---

> ### Author Response · Authors · 2022-11-19
> **Responses to Reviewer PPSy**
>
> We sincerely appreciate the insightful comments from the reviewer. Below lists our responses to the questions raised by the reviewer.
>
> Q1: The architecture is not the only factor affecting adversarial transferability.
>
> A: It is reasonable that modifying parameters will change the ranking. For example, a model trained by adversarial training methods can surely obtain higher adversarial robustness than a model trained by normal strategy. While we think it is still significant to apply DeepGuiser because whatever method we use to train the parameters, transferability is an essential property of the neural architecture. We'd say that under the same training configuration, two models with the same architecture while with different parameters tend to have higher transferability.
>
> Q2: The proposed method largely increases the inference cost. Trade-off for balancing FLOPS & security.
>
> A: Actually, we found that it does not always stand that greater changes in architecture will decrease the transferability more. We add several additional experiments to evaluate FLOPS versus adversarial accuracy reward. We sample several architecture disguising decisions from DeepGuiser and select out the architectures with different FLOPS budget to test their transferability, As can be seen in the following table. In some cases, paying more FLOPS will result in higher reward, but it does not keep going throughout the curve.
>
> ---------
>
>
> Arch            |     FLOPS ratio
>
> ResNet      |     1.22  |  1.68  | 2.40  | 2.78  | 3.32
>
> Reward      |      27.95  | 25.94  | 35.11  | 37.74  | 27.17
>
> VGGNet    | 1.15 | 1.58  | 2.00 | 2.21 | 2.61
>
> Reward | 28.05 | 22.39 | 23.44 | 29.49 | 23.53
>
> MobileNetV2 | 1.26 |  2.13 | 3.43 | 4.34 | 5.39
>
> Reward | 10.08  | 6.99 | 14.63 | 12.25 | 17.34
>
> ----------------------------
>
> We will make more exploration on this point, to discuss if we can provide a convenient way for the users to decide the disguising option based on their computation budget, such that they can trade-off the security level and the disguising overhead.
>
> Q3: Issues about clean accuracy.
>
> A: Acc_clean stands for clean test accuracy. We will add the notation. The accuracy is normal as we only set the base channel number as 20. Lower accuracy of disguised architecture means the neural architecture is not as good as the original architecture, thus even if the attackers reverse the fake network architecture, they get fewer benefits from the attacking. The bolding means best result under corresponding metrics with different disguising methods. Some of them are higher better, while some of them are lower better. There might be some mistakes, thank you for pointing out that and we will fix it.

---

> > ### Comment · Reviewer_PPSy · 2022-11-24
> > **Thanks for your response**
> >
> > I would like to thank the authors for providing the rebuttal. My concerns about Q2 and Q3 are solved. But in Q1, I do not mean adversarial training. I mean training the same architecture DNN with different random seeds. TransAdvBench is useful only if randomness does not influence the transferability much. Empirical verification is necessary to support the claim "under the same training configuration, two models with the same architecture while with different parameters tend to have higher transferability."
> >
> > As far as I can see, the authors make lots of promises instead of modifying the manuscript. In the last paragraph of "Responses to Reviewer moEm", the authors even forget to add the specific reference of the review, seeing "in the responses to Q5 of Reviewer and please ...". I think the manuscript is not ready for publication despite its novelty.

---

### Official Review · Reviewer_moEm · 2022-10-24

**Confidence:** 3
**Correctness:** 2
**Technical Novelty And Significance:** 3
**Empirical Novelty And Significance:** 2
**Recommendation:** 3

**Clarity, Quality, Novelty And Reproducibility:**

-[clarity] the task in this paper is clearly stated (disguise attacks through side channel). The task is of significance in terms of AI security.

-[clarity] fig2 (b) and (c) are clear examples on functionally equivalent replacement of an operation. This helps the reader better understand the section 4.1.

-[quality] important relevant works are comprehensively reviewed in section 2.

-[novelty] the idea of finding functional equivalent to defend against side-channel attack is novel. I think this method would work, but the evaulations are not confusing since they are indirect evaluation, and the side-channel attack evaluation is left unrevealed.

-[reproducibility] there is python code.

**Strength And Weaknesses:**

Strengths

-[motivation] the topic is of significance in terms of AI security. the disadvantages of existing methods are clearly discussed.

-[design] the design of searching functionally equivalent architectures to defend against side-channel attacks is interesting.

-[experiments] the proposed induces less overhead compared to previous methods.

Weaknesses

-[clarity, major] I find Eq.3 confusing. Since A and B are functional equivalents, the output of A with an adversarial example created from B should produce exactly the same output, making the same error if there is any. Since changing different disguises would not change the fact that A/B are functionally equivalent, the loss degenerates into a constant (the function output cannot be changed in the disguise space). And, how is the problem formulation in section 4.1 relevant to side-channel attacks?

-[clarity, minor] for functional equivalent operations such as fig2.(b) (c), are the padded zeroes frozen during the training process? If not, the disguise cannot be claimed as functionally equivalent.

-[evaluation, major] The evaluation of disguise architecture is not very straightforward. The evaluation is not directly based on neural network architecture extraction, but is based on the transferability attack after the extraction. Since the disguise model is functionally equivalent, an example created using model B should lead to the same output in model A. But in the experiments, the models are not equivalent due to re-initialized parameters, which makes the models not equivalent to each other. Then, that means the effectiveness of the proposed method is tightly entangled with randomly initialized model parameters. We don't know whether the transfer attack success rate stems from such parameter differences -- because (emphasize again) when A and B are the same mapping, they produce the same result and will not make a difference. In that case, does that mean what the model has learned is to some extent the differences in different parameter initializations instead of good disguise architecture choice? What if we fix the model parameter and convert it to a series of mathematically equivalent functions? The claim/practice inconsistency makes the evaluation confusing and not convincing. Apart from that, is the proposed defense really effective against side-channel attack itself?

-[suggestion] it is suggested to discuss adaptive attack where the attacker has full knowledge of the disguise algorithm.

**Summary Of The Paper:**

This paper presents a disguise architecture method to search for a better functionally-equivalent architecture that leads to a lower transferability attack success rate even if the model is leaked through side-channel attack. To solve the disguise architecture search problem, a reward function is built for evaluating network architectures, and a policy learning method is proposed to solve the architecture search based on the reward. Expeirments are conducted on cifar-10, cifar-100, and tiny-imagenet.

**Summary Of The Review:**


This paper aims to defend against side-channel attack, but the evaluation is based on the transferability attack, with the side-channel attack effectiveness hidden within the transferability attack, which makes the evaluation obscure. Although the idea of finding a functional equivalent is very intersting, the inconsistency where the functional equivalence is actually not guaranteed in the experiments further make the method and evaluation confusing and less convincing -- it's is hard to attribute the effectiveness to the disguise architecture or parameter difference. There is still a room for improvement in terms of clarity and evaluation. I'd recommend weak reject currently.

---

> ### Author Response · Authors · 2022-11-19
> **Responses to Reviewer moEm**
>
> We appreciate the reviewer's time and efforts on reviewing our paper, and giving us thorough suggestions for helping us improve this work. Many of the ideas have greatly inspired us. The following lists our responses and hope to have a discussion with you.
>
> Q1: The clarity about Eq.3 and side-channel attacks.
>
> A: We are sorry that the unclear expression leads to some misunderstanding. The B in Eq.3 is another model trained by the attacker, and it does not share weight parameters with A. The assumption is that an attacker steals A's architecture but he gets architecture B (as we have disguised A to B). Then he trains surrogate model B to attack A by himself. Therefore, in Eq.3, B is not functionally equal to A. In other words, A in Eq.3 is functionally equal to a deployed B'. B' shares the parameters of A while it has a different architecture than A. And what the attack gets is the arch of B'. While B is trained by the attack and has totally different parameters as B'.
>
> Side-channel attack is one of the most common ways to hack the AI system and help the attackers to reveal the neural architectures. Our formulation is not only designed for side-channel attacks. Any neural architecture reversing attacks can be defended by DeepGuiser.
>
> Q2: Clarity about the op transformation.
>
> A: It does not happen in the training process. It happens after the training process. The operation transformations are performed in a post-training manner. Before deploying a model A, we will transform its operations to make the execution look the same as another architecture. Note that the transformation will not change the numerical computation results. For example, transforming Conv 1x1 to Conv3x3 by padding zeros will have no impact on the results. Therefore, the disguising can ensure functional equivalence.
>
> Q3: Evaluation about the disguised architecture.
>
> A: As explained in Q1, B in Eq.3 is not functionally equivalent to A because the attackers cannot obtain the weight parameters and they need to retrain the surrogate model to generate adversarial examples for transfer attack A. Moreover, the attacker cannot directly generate an adversarial example based on the disguised A. Our evaluation decouples the parameter initialization and the disguiser learning process, as we assume all the involved models are trained independently with the same training configurations. The proposed defense can work against side-channel attack because such attacks often snoop on the execution hints and traces. And DeepGuiser can change the execution behavior, such that it makes the snooping ineffective.
>
> Q4: Discussion on adaptive attack.
>
> A: Part of the question has been answered in the responses to Q5 of Reviewer and please refer to that. In addition, we think query-based adaptive attack may pose new threats to our method as it only cares about the responses from the network. And DeepGuiser will not change the network's outputs/responses because of the functional equivalence. We will further discuss that in our revision.

---

> > ### Comment · Reviewer_moEm · 2022-11-30
> > **Thanks for you response**
> >
> > The response addressed some of my concerns, and I think the clarity of this manuscript indeed has to be improved to reduce the risk of misunderstanding for future readers. In the response to my Q4, the reviewer reference is missing -- I did not find a response to a "Q5" that is very relevant to adaptive attack...

---

### Official Review · Reviewer_DYHG · 2022-10-31

**Confidence:** 4
**Correctness:** 3
**Technical Novelty And Significance:** 3
**Empirical Novelty And Significance:** 3
**Recommendation:** 6

**Clarity, Quality, Novelty And Reproducibility:**

Clarity: Above average.
Quality: Above average.
Novelty: Discussing a new problem but the solution is not new.
Reproducibility: Not sure.

**Details Of Ethics Concerns:**

No Ethics Concerns

**Strength And Weaknesses:**

 1. Strength:
- The problem is new and interesting.

2. Weakness:
- The overhead is huge. I am not sure if the method is applicable to other types of models (discussed in the questions).
- The solutions are not new. Predictor+RL are very standard methods in NAS. Predicting adversarial accuracy is hard.

**Summary Of The Paper:**

This paper proposes a model disguising method. The key motivation is that: when the model architecture is obtained by the attacker, they can rebuild the weight parameters through a transfer learning attack. Then the attacker will generate adversarial examples exploiting the rebuilt model.

 By smartly adding some redundant operations (through RL), the paper can hide the model architectures while increasing the adversarial accuracy when being attacked.


**Summary Of The Review:**

1) You claim that your disguised models are quality neutral in terms of model accuracy against the original models. Is there any evaluation to prove it?

2) If the attacker is aware of this protection method, they can simply conduct random pruning (or NAS) to find the DNN architecture in the reduced search space.

3) Also, in my opinion, this method can only be applied to NASNet or similar models. I don't think it is applicable to recent advances like EfficientNet or ViT. That is because these layers models are wired in sequence, the attacker would simply remove your disguised modules
 before starting the transfer learning.

Overall, I think the paper proposes a very interesting idea. We need to have two versions of our models; the original model and another disguised model for deployment purposes.  However, the paper does not dig into the problem enough. There are many easy ways to neutralize this defense mechanism. At least, none of them are discussed.

Minor: Acc_clean is not explained anywhere.

---

> ### Author Response · Authors · 2022-11-19
> **Responses to Reviewer DYHG**
>
> We sincerely appreciate the reviewer's time and efforts on reviewing our paper, and giving us insightful comments and suggestions. Below lists our responses and discussions. Hope that the responses can address your concerns.
>
> Q1: The overhead is huge.
>
> A: DeepGuiser will indeed introduce addition overhead on model size and inference latency, as we have presented in the paper (around 10% ~40% extra latency). While in most cases, we have to pay certain overhead for obtaining a higher security level. For example, a representative approach that can hide memory access behavior, ORAM, has a communication lower bound of O(log(n)), which means we need to pay O(log(n)) times more overhead on communication. Overall, we seek to minimize the overhead, but it is unavoidable. And we think current overhead is within the acceptance range.
>
> Q2: The solutions are not new and predicting adv acc is hard.
>
> A: We'd like to share our opinion on that. Predictor and RL are frequently applied in NAS literature, while our work is the first to integrate them as an automatic disguising method for protecting neural arch confidentiality.  It is not a simple combination. As you have mentioned, predicting adversarial accuracy is more difficult than predicting individual model accuracy as it involves a pair of architectures. Therefore, to achieve a good result, there are many problems that should be addressed, e.g. the collection of TransAdvBench, the architecture encoder design, the RL algorithm design and so on.
>
> Q3: If disguised models are quality neutral in terms of model accuracy against the original models.
>
> A: We understand the "quality neutral" as that "the accuracy of disguised model will maintain as the original models". If there is no misunderstanding, the answer would be YES. The disguising model can theoretically ensure exactly the same functionality as the original model. As the example shown in Fig.2(b-c), when padding surrounding zeros in a 1x1 convolution kernel to disguise to a 3x3 kernel, the numerical results will keep unchanged.
>
> Q4: If the attacker is aware of this protection method, they can simply conduct random pruning (or NAS) to find the DNN architecture in the reduced search space.
>
> A: We think it is possible. If the attacks are aware of the DeepGuiser method, they can leverage the same mechanism as DeepGuiser to reverse-search the original model in a reduced search space. For example, someone sees an operator conv3x3, he can infer that the original operator must be one of conv3x3, conv1x1, skip connection and null. Then by constructing or leveraging TransAdvBench to train his own "reverse-disguiser", they can try to crack the disguising.
>
> However, we think the cracking process is not easy because of the following two reasons. First, if we disguise an original arch A to another arch B, while an attacker may infer another arch C from B, both A and C are low transferable to B. In this case, C is not equal to A and maybe also with low transferability to A, thus the cracking will fail. Second, though the search space is reduced, the search space is still large as there are still a lot of valid transformations, which will make the cracking not that easy. Nevertheless, we agree that it is an important point that we should consider, and we will deeply discuss that in our next revision.
>
> Q5: if it is applicable to recent advances like EfficientNet or ViT.
>
> A: That is a really good point that we missed. We haven't considered the network architectures outside conventional CNNs, e.g. ViT or LSTM-like sequential models. We now have no experimental results to prove that our method can always work in these cases but we think they have some essential similarity. It is worth further exploration.

---

### Decision · Program_Chairs · 2023-01-20

**Decision:**

Reject

**Justification For Why Not Higher Score:**

The reviewers have significant concerns about this paper and do not believe it is ready for publication at this time.

**Justification For Why Not Lower Score:**

N/A

**Metareview: Summary, Strengths And Weaknesses:**

This paper presents a novel mechanism for defending against adversarial transfer attacks by converting a neural network into a functionally equivalent version with a different architecture. Although the reviewers found the paper to be novel and interesting, they raised several major concerns, including the clarity of the proposed method, insufficient empirical evaluations, and significant overhead. The authors' rebuttal addressed these concerns only partially, and the reviewers were still concerned about the missing empirical evaluations and the need for further polishing of the writing.

The authors are encouraged to carefully address these concerns and make a stronger submission next time.